# Nonsmoking after simultaneous alcohol abstinence and smoking cessation program was associated with better drinking status outcome in Japanese alcohol-dependent men: A prospective follow-up study

Akira Yokoyama[1]*, Mitsuru Kimura[1], Atsushi Yoshimura[2], Sachio Matsushita[1], Junichi Yoneda[1], Hitoshi Maesato[1], Yasunobu Komoto[3], Hideki Nakayama[4], Hiroshi Sakuma[5], Yosuke Yumoto[1], Tsuyoshi Takimura[1], Tomomi Toyama[1], Chie Iwahara[6], Takeshi Mizukami[1], Tetsuji Yokoyama[7], Susumu Higuchi[1]

1 National Hospital Organization Kurihama Medical and Addiction Center, Yokosuka, Kanagawa, Japan, 2 Division of Psychiatry, Tohoku Medical and Pharmaceutical University, Sendai, Miyagi, Japan, 3 Department of Psychiatry, Yoshino Hospital, Machida, Tokyo, Japan, 4 Department of Psychiatry, Asahiyama Hospital, Sapporo, Hokkaido, Japan, 5 Department of Psychiatry, National Hospital Organization Saigata Medical Center, Joetsu, Niigata, Japan, 6 Department of Psychiatry, Hakuhou Clinic, Saitama, Saitama, Japan, 7 Department of Health Promotion, National Institute of Public Health, Wako, Saitama, Japan

* a-yoko@ab.auone-net.jp

## Abstract

### Background

Growing evidence suggests that intervention for smoking cessation enhances alcohol abstinence in treatment settings for alcohol dependence. However, research in this field is rare in Asians.

### Method

We prospectively investigated the association of smoking status with drinking status using 9 surveys mailed during a 12-month period in 198 Japanese alcohol-dependent men (70 never/ex-smokers and 128 smokers) who admitted for the first time and completed a 3-month inpatient program for simultaneous alcohol abstinence and smoking cessation.

### Results

Nonsmoking during the first month after discharge and at the end of follow-up was reported in 28.9% and 25.0% of the baseline smokers, respectively. Kaplan-Meier estimates showed that a 12-month alcohol abstinence and heavy-drinking-free status were more frequent among never/ex-smokers (45.1% and 59.8%, respectively) and baseline smokers who quit smoking during the first month after discharge (59.0% and 60.8%, respectively), compared with sustained smokers (30.0% and 41.2%, respectively). Among the baseline smokers, the multivariate odds ratio (95% confidence interval) for smoking cessation during the first

**Data Availability Statement:** All relevant data are within the paper and its Supporting Information files.

**Funding:** This research was supported by Health Labour Sciences Research Grant, Japan (H26-seishin-ippan-006) and Intramural Research Grant for Neurological and Psychiatric Disorders of National Center of Neurology and Psychiatry (25-2). The funders had no role in study design, data collection and analysis, decision to publish, or preparation of the manuscript.

**Competing interests:** The authors have declared that no competing interests exist.

month were 2.77 (1.01–7.61) for alcohol abstinence during the period and 2.50 (1.00–6.25) for use of varenicline, a smoking cessation agent, during the inpatient program. After adjusting for age, drinking profile, lifestyle, family history of heavy or problem drinking, lifetime episodes of other major psychiatric disorders, and medications at discharge, the multivariate hazard ratios (HRs) for drinking lapse were 0.57 (0.37–0.89) for the never/ex-smoking and 0.41 (0.23–0.75) for new smoking cessation groups, respectively, compared with sustained smoking, while the corresponding HRs for heavy-drinking lapse were 0.55 (0.33–0.90) and 0.47 (0.25–0.88), respectively. The HR for drinking lapse was 0.63 (0.42–0.95) for the nonsmoking group (vs. smoking) during the observation period, while the HR for heavy-drinking lapse was 0.58 (0.37–0.91) for the nonsmoking group (vs. smoking) during the observation period. Other significant variables that worsened drinking outcomes were higher daily alcohol intake prior to hospitalization, family history of heavy or problem drinking and psychiatric medications at discharge.

## Conclusion

Nonsmoking was associated with better outcomes on the drinking status of Japanese alcohol-dependent men, and a smoking cessation program may be recommended to be integrated into alcohol abstinence programs.

## Introduction

Alcohol-dependent individuals have a high prevalence of smoking and/or nicotine dependence [1–3], and the combination yield more serious problems in diverse mental and physical areas [4–6]. Growing evidence suggests that intervention for smoking cessation enhances, rather than jeopardizes, alcohol abstinence in treatment settings for alcohol dependence [2, 3]; for example, a meta-analysis of 19 randomized controlled trials showed that smoking cessation interventions provided during addictions treatment were associated with a 25% increased likelihood of a more than 6-month abstinence from alcohol and illicit drugs [7]. In addition, prospective data after substance use treatment showed that stopping smoking during the first year after treatment predicted better long-term substance use outcomes over a 9-year period [8].

The Kurihama Medical and Addiction Center has provided an inpatient program for simultaneous alcohol abstinence and smoking cessation since 2010, and we previously reported that Japanese alcohol-dependent men (73% were current smokers) were highly motivated to stop smoking during the program; furthermore, half of the totally or mostly abstinent subjects during a 6-month period after program completion were never/ex-smokers or new tobacco quitters [9]. However, a prospective evaluation of the drinking status after simultaneous drinking and smoking interventions for alcohol-dependent patients has not been conducted in an Asian population. We are hypothesizing that nonsmoking would positively impact on drinking outcome in an Asian population.

In the present study, we prospectively investigated the association of never/ex-smoking and new smoking cessation with drinking status using 9 questionnaires mailed during a 12-month follow-up period in alcohol-dependent men who had completed a simultaneous alcohol abstinence and smoking cessation program.

## Materials and methods

### Subjects

This study was designed as part of a prospective drinking outcome study [10] in patients who had been admitted to the Center for the treatment of alcohol dependence between January and December 2014. Five hundred and forty-three men (20–85 years old) were admitted for the treatment of alcohol dependence during the study period and provided written informed consent to participate in the original prospective study.

The standard treatment in the Center consisted of a 3-month inpatient alcoholism-rehabilitation program (ARP). The 3-month ARP included medical treatment for withdrawal and physical problems, an educational program, group cognitive behavioral therapy, and participation in self-help group meetings. The subjects were recommended to receive regular outpatient treatment at our hospital or another clinic in the subject's neighborhood and to attend self-help group meetings after discharge.

Excluding subjects who left the hospital after short-term detoxification or during the ARP, 354 subjects completed the 3-month ARP. Excluding subjects who had participated in the ARP on two or more occasions, 242 subjects participated in the ARP for the first time. Since the prognosis for sobriety was very poor for the patients in multiple inpatient treatment [10], only first-time treatment patients were included in the present study. Mailed questionnaires could not be sent to 8 subjects after hospital discharge because of an unknown address or death within a month after discharge. Out of the remaining 234 subjects, 198 subjects responded to the mailed questionnaire and were included in the present study's analysis.

Structured questionnaires asking the first date of drinking lapse, the first date of heavy-drinking lapse [$\geq$60 g ethanol (6 drinks) per day], and smoking status were mailed every month until 6 months and every 2 months from 6 to 12 months after discharge. The subjects were asked to complete and return the questionnaires. When the subjects neglected to respond, our technical staff phoned the subjects and asked them to return the questionnaires. When a subject did not respond to three consecutive questionnaires, we stopped sending him questionnaires and the subject was handled as a dropout at the last reply. The subjects were compensated with a 500 JPY ($\approx$ 4.1 USD, 2015) prepaid card per one reply. All the subjects were diagnosed with alcohol dependence based on the assessment by psychiatrists using the ICD-10 criteria [11]. The ethics committee of the Kurihama Medical and Addiction Center reviewed and approved the proposed study (No. 188), and all procedures involved in this study were performed in accordance with the Declaration of Helsinki with written informed consent from each participant.

### Baseline clinical characteristics

Information on age at the time of the first drinking, age at the start of regular drinking, usual alcohol consumption, smoking status, lifestyle, and family history of heavy or problem drinking among grandparents, parents, and siblings was obtained from the subjects by trained interviewers using a structured questionnaire and, when available, from their significant others. Usual alcohol consumption during the preceding year was expressed in grams of ethanol per day, based on the amount and ethanol concentration of alcoholic beverages consumed. Lifetime episodes of other major psychiatric disorders were investigated using Mini-International Neuropsychiatric interview (MINI) Japanese version by certified clinical psychologist [12].

### Smoking cessation program [9]

The program was conducted under an organization-wide smoke-free environment (i.e., no smoking on hospital premises for both the subjects and medical stuff). Varenicline, a smoking cessation agent, was recommended for current smokers during the 3-month hospital stay, and 61 (47.7%) of the 128 current smokers selected its use, with dosage titration to 2 mg per day during the first week. All the subjects attended four 40-min group study meetings regardless of their smoking status. The themes of the study meetings were 1) nicotine dependence and coping skills for smoking cessation, 2) tobacco-related cancers and chronic obstructive pulmonary disease, 3) smoking and various arteriosclerotic diseases, and 4) smoking and life style diseases. The effects of smoking cessation on improving long-term alcohol abstinence as well as mental and physical recovery were highlighted by providing research evidence at every meeting. In collaboration with psychiatrists, physicians advised the subjects of the merits of smoking cessation based on selected medical examinations, such as lung function tests using spirometry and smoking-related cancer screening.

### Statistical analysis

The rates of alcohol abstinence and the heavy-drinking-free status were estimated by the Kaplan-Meier method, and differences between groups were analyzed using the log-rank test. Multiple logistic regression analysis was used to estimate the multivariate odds ratios (ORs). The multivariate hazard ratios (HRs) of selected variables on the first drinking lapse and the first heavy-drinking lapse were assessed using the multivariate Cox proportional hazards model. To assess the effects of 18 subjects who dropped out of the study during the study period, sensitivity analyses were conducted with the assumption that all the dropout subjects drank or drank heavily at the time of dropout or that all the dropout subjects continued to abstain from alcohol or were heavy-drinking free. Sensitivity analyses was also performed using the first-time and multiple treatment patients. Among the baseline smokers, the number of cigarettes/day at the baseline was compared with those after one month and at the end of the follow-up by Wilcoxon signed-rank test. All analyses were performed using the SAS statistical package (version 9.4; SAS Institute, Cary, NC).

### Results

Table 1 shows characteristic of 198 subjects who were followed and 44 subjects who were not followed out of 242 first-time hospitalized patients who completed the 3-month inpatient program. The follow-up subjects were older than the non-tracking subjects, but the other backgrounds, including drinking and smoking habits, lifestyle, family history of heavy or problem drinking, lifetime episodes of other psychiatric disorders, and medications at discharge did not differ between the two groups. The interview using MINI revealed lifetime episodes of major psychiatric disorders in 58 (24.0%); 40 had affective disorder, 29 had anxiety disorder, 6 had antisocial personality disorder, 4 had psychotic disorder, 1 had drug dependence, and 1 had eating disorder. At discharge, 91 (37.6%) were taking medications for sobriety; acamprosate in 73 and disulfiram in 32, and 136 (56.2%) were taking other psychiatric medications; sleeping drugs in 120, antidepressants in 44, antipsychotics in 26, and antianxiety drugs in 24.

The Kaplan-Meier estimates for the proportions of subjects who continued alcohol abstinence (left) and a heavy-drinking-free status (right) are shown according to their smoking status during the first month after discharge in Fig 1. Thirty-seven (28.9%) of the 128 baseline smokers (current smokers) quit smoking during the first month. Alcohol abstinence and a heavy-drinking-free status were observed more frequently in never/ex-smokers (45.1% and

**Table 1. Background characteristics in the alcohol-dependent men who had completed 3-month inpatient alcoholism-rehabilitation program.**

| | Referent Subjects | | |
|---|---|---|---|
| | followed n = 198 | unfollowed n = 44 | P |
| Age (yrs) | 57.0±12.8 | 52.2±10.8 | 0.012 |
| Usual alcohol intake (g ethanol/d) | 115.7±60.8 | 125.8±76.1 | 0.41 |
| Alcoholic beverage most frequently consumed | | | |
| beer/low-malt beer (4%-5% ethanol v/v) | 14.6% | 11.4% | 0.81 |
| canned chuhai (4–9%) | 11.6% | 18.2% | |
| wine (12%) | 2.5% | 2.3% | |
| sake (15%-16%) | 13.6% | 13.6% | |
| shochu (20%-25%) | 44.9% | 38.6% | |
| whyskey/other spirits (40%) | 12.6% | 15.9% | |
| Age at first drinking (yrs) | 17.6±3.0 | 17.0±3.0 | 0.30 |
| Age at the start of regular drinking (yrs) | 24.2±7.7 | 24.1±8.5 | 0.93 |
| Cigarette smoking | | | |
| Never smoking | 13.1% | 13.6% | 0.77 |
| Current smoking | | | |
| 1–19 cigs/d | 25.8% | 31.8% | |
| 20+ cigs/d | 38.9% | 38.6% | |
| (Use of Varenicline at the hospital stay)* | (30.8%) | (29.5%) | |
| Ex-smoking | 22.2% | 15.9% | |
| Living alone | 29.3% | 38.6% | 0.28 |
| Unemployed | 61.1% | 50.0% | 0.18 |
| Family history of heavy/problem drinking | 44.9% | 52.3% | 0.41 |
| Lifetime episodes of other psychiatric disorders | 21.7% | 34.1% | 0.12 |
| Medication at discharge | | | |
| Acamprosate or Disulfiram | 39.9% | 27.3% | 0.13 |
| Other psychiatric medication | 55.1% | 61.4% | 0.50 |

*Percentage values are those in the current smokers.

Data were expressed by mean±SD or percentage values for column.

P values are by Fisher's exact test for percentage values or Student t-test for continuous variables.

59.8%, respectively) and baseline smokers who had quit smoking (59.0% and 60.8%, respectively) than in baseline smokers who continued smoking (30.0% and 41.2%, respectively).

The sensitivity analysis for Fig 1 with the assumption that all the dropout subjects drank or drank heavily at the time of dropout showed essentially unchanged or enhanced results (S1 Fig). Meanwhile, the sensitivity analysis with the assumption that all the dropout subjects continued to abstain from alcohol or were heavy-drinking free showed slightly weakened but essentially unchanged results.

Patients receiving multiple treatments were excluded due to their very poor sobriety prognosis. Of these, 74 responded to the questionnaire, and their 1-year abstinence rate was 23.1% ± 5.3% (mean ± SE), while the abstinence rate for the 198 first-time treatment patients was 41.0% ± 3.6%. A sensitivity analysis was also performed for Fig 1 using a total of 272 first-time and multiple treatment patients (S2 Fig). Although the overall abstinence rate decreased, the results of the sensitivity analysis were essentially unchanged.

Table 2 compared the characteristics between the 37 baseline smokers who did not smoke within 1 month after discharge and the 91 baseline smokers who smoked during that period. A multiple logistic regression analysis showed that the multivariate OR (95% confidence interval;

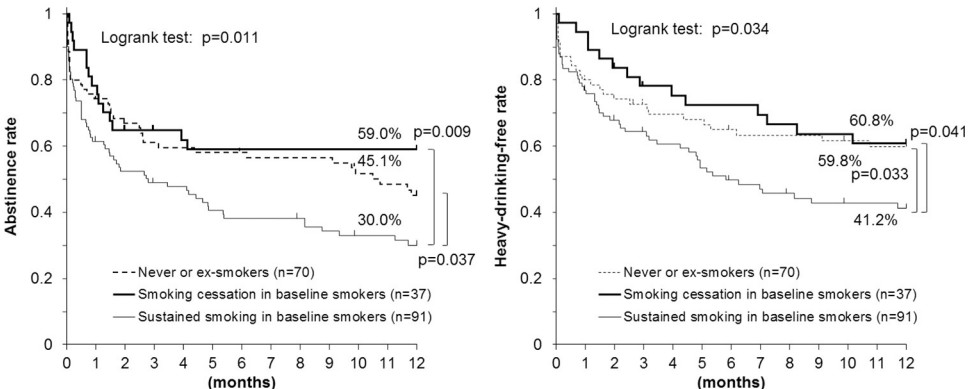

**Fig 1. Smoking status within 1 month after discharge and drinking status after a 3-month inpatient program for alcohol abstinence and smoking cessation in Japanese alcohol-dependent men.** Thirty-seven (28.9%) of the 128 baseline smokers quit smoking during the first month. Alcohol abstinence (left) and a heavy-drinking-free status (right) were more frequent among never or ex-smokers and baseline smokers who quit smoking, compared with sustained smokers.

**Table 2. Nonsmoking within 1 month after inpatient program in baseline smokers and its associated factors in Japanese alcohol-dependent men.**

| | Smoking status in baseline smokers within 1 month after discharge | | | |
|---|---|---|---|---|
| | Nonsmoking n = 37 | Smoking n = 91 | P | Multivariate OR (95%CI) |
| Age (yrs, mean±SD; per +10 yrs) | 53.4±12.1 | 54.3±11.0 | 0.66 | 0.96 (0.63–1.47) |
| Age at first drinking (yrs) | | | | |
| ≤15 yrs | 21.6% | 24.2% | 0.84 | 0.91 (0.21–3.89) |
| 16–19 yrs | 48.6% | 51.6% | | 0.78 (0.27–2.27) |
| ≥20 yrs | 29.7% | 24.2% | | referent |
| Age at the start of regular drinking (yrs) | | | | |
| ≤19 yrs | 16.2% | 19.8% | 0.87 | 0.95 (0.23–3.97) |
| 20–25 yrs | 56.8% | 50.5% | | 1.49 (0.54–4.14) |
| ≥26 yrs | 27.0% | 29.7% | | referent |
| Usual alcohol intake (g ethanol/d, mean±SE; per +22 g/d) | 109.4±9.2 | 123.3±6.5 | 0.24 | 0.87 (0.73–1.05) |
| Alcoholic beverage most frequently consumed | | | | |
| beer/canned chuhai (4–9%) | 32.4% | 29.7% | 0.71 | 1.72 (0.60–4.93) |
| sake/wine (12–16%) | 13.5% | 9.9% | | 2.07 (0.51–8.37) |
| shochu/whiskey/other spirits (20–40%) | 54.1% | 60.4% | | referent |
| Alcohol abstinence within 1 month after discharge | 78.4% | 61.5% | 0.098 | 2.77 (1.01–7.61) |
| Baseline current smoking (cigs/d, mean±SE; per +20 cig/d) | 19.7±1.7 | 19.0±1.1 | 0.71 | 1.59 (0.64–3.96) |
| Use of varenicline at the hospital stay | 56.8% | 44.0% | 0.24 | 2.50 (1.00–6.25) |
| Living alone | 67.6% | 68.1% | 1 | 1.01 (0.40–2.56) |
| Unemployed | 40.5% | 48.4% | 0.44 | 2.18 (0.84–5.65) |
| Family history of heavy/problem drinking | 48.6% | 41.8% | 0.56 | 1.37 (0.57–3.32) |
| Lifetime episodes of other psychiatric disorders | 24.3% | 19.8% | 0.63 | 1.22 (0.43–3.49) |
| Medication at discharge | | | | |
| Acamprosate or disulfiram | 51.4% | 39.6% | 0.24 | 2.00 (0.79–5.05) |
| Other psychiatric medication | 59.5% | 60.4% | 1 | |

Data were expressed by mean±SD/SE or percentage values for column.

P values are by Fisher's exact test for percentage values or Student t-test for continuous variables.

Multivariate odds ratio (OR) and 95% confidence interval (CI) for non-smoking by the logistic regression model, where all variables were simultaneously entered into the model

**Table 3. Drinking status during 12 months after inpatient program and its determinants including smoking status within 1 month in Japanese alcohol-dependent men.**

| | Drinking relapse | Heavy drinking |
|---|---|---|
| | Multivariate HR (95% CI)* | Multivariate HR (95% CI)* |
| Age (per +10 yrs) | 1.07 (0.90–1.28) | 1.05 (0.86–1.29) |
| Age at first drinking (yrs) | | |
| ≤15 yrs | 1.28 (0.70–2.34) | 1.73 (0.87–3.46) |
| 16–19 yrs | 0.92 (0.56–1.50) | 1.20 (0.68–2.13) |
| ≥20 yrs | referent | referent |
| Age at the start of regular drinking (yrs) | | |
| ≤19 yrs | 1.06 (0.58–1.92) | 1.11 (0.57–2.16) |
| 20–25 yrs | 1.42 (0.88–2.29) | 1.62 (0.93–2.83) |
| ≥26 yrs | referent | referent |
| Usual alcohol intake (per +22 g/d) | 1.08 (1.01–1.16) | 1.09 (1.01–1.17) |
| Alcoholic beverage most frequently consumed | | |
| beer/canned chuhai (4–9%) | 0.98 (0.62–1.55) | 0.83 (0.49–1.42) |
| sake/wine (12–16%) | 1.54 (0.89–2.67) | 1.61 (0.88–2.93) |
| shochu/whiskey/other spirits (20–40%) | referent | referent |
| Smoking status within 1 month after discharge | | |
| Never or ex-smokers | 0.57 (0.37–0.89) | 0.55 (0.33–0.90) |
| Smoking cessation in baseline smokers | 0.41 (0.23–0.75) | 0.47 (0.25–0.88) |
| Smoking in baseline smokers | referent | referent |
| Living alone | 0.83 (0.55–1.27) | 0.93 (0.58–1.47) |
| Unemployed | 1.19 (0.78–1.82) | 1.33 (0.83–2.12) |
| Family history of heavy/problem drinking | 1.56 (1.06–2.31) | 1.34 (0.87–2.07) |
| Lifetime episodes of other psychiatric disorders | 1.16 (0.73–1.85) | 1.22 (0.72–2.05) |
| Medication at discharge | | |
| Acamprosate or disulfiram | 1.06 (0.71–1.59) | 1.04 (0.66–1.64) |
| Other psychiatric medication | 1.57 (1.05–2.35) | 1.60 (1.02–2.52) |

Multivariate hazard ratio (HR) and 95% confidence interval (CI) by the Cox proportional hazards model, where all variables were simultaneously entered into the model.

CI) for smoking cessation was 2.50 (1.00–6.25) for those who had used varenicline during hospitalization and 2.77 (1.01–7.61) for those who did not drink with 1 month after discharge.

A Cox proportional hazards analysis (Table 3) showed that the multivariate HRs (95% CI) for drinking lapse were 0.57 (0.37–0.89) and 0.41 (0.23–0.75) in never/ex-smokers and baseline smokers who quit smoking, respectively, compared with baseline smokers who continued to smoke, while the HRs for heavy-drinking lapse were 0.55 (0.33–0.90) and 0.47 (0.25–0.88), respectively. Higher daily alcohol intake and taking psychiatric medications at the time of discharge were associated with higher HRs for both drinking lapse and heavy-drinking lapse. Family history of heavy or problem drinking was associated with higher HRs for drinking lapse.

Fig 2 shows the Kaplan-Meier estimates for the proportions of subjects who continued alcohol abstinence (left) or a heavy-drinking-free status (right) according to whether the subjects have smoked prior to or at the time of either the first drinking episode or the end of follow-up without it. Ninety-five (48.0%; 69 baseline never/ex-smokers and 26 baseline smokers) of the 198 subjects did not smoke prior to and at the time of either the first drinking lapse or the end of follow-up without it. One baseline ex-smoker smoked at the time of the first drinking

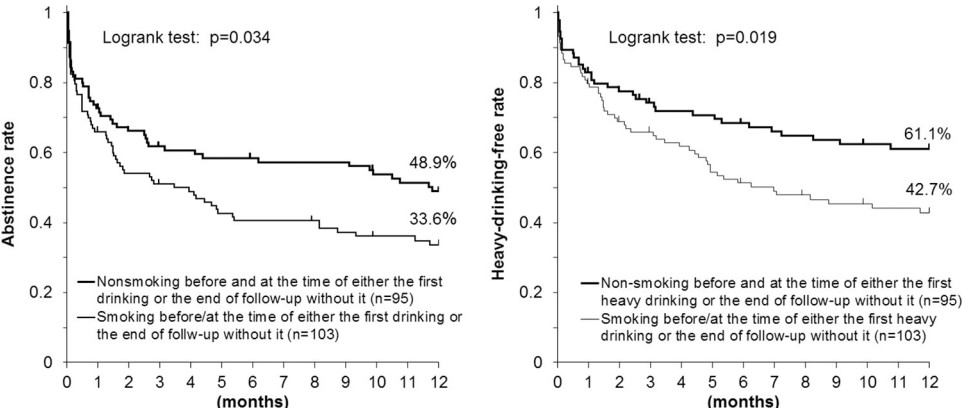

**Fig 2. Smoking status and drinking status after a 3-month inpatient program for alcohol abstinence and smoking cessation in Japanese alcohol-dependent men.** Alcohol abstinence (left) and a heavy-drinking-free status (right) were more frequent in nonsmoking subjects, compared with subjects who reported smoking prior to or at the time of either their first drinking or heavy drinking episode or at the end of the follow-up without the episode.

episode. Alcohol abstinence and a heavy-drinking-free status were more frequently observed among nonsmoking subjects (48.9% and 61.1%, respectively), compared with smokers (33.6% and 42.7%, respectively).

The sensitivity analysis for Fig 2 with the assumption that all the dropout subjects drank or drank heavily at the time of dropout showed essentially unchanged or enhanced results (S3 Fig). Meanwhile, the sensitivity analysis with the assumption that all the dropout subjects continued to abstain from alcohol or were heavy-drinking free showed slightly weakened but essentially unchanged results.

Another sensitivity analysis for Fig 2 was performed using the 272 first-time and multiple treatment patients described above (S4 Fig). Although the overall abstinence rate decreased, the sensitivity analysis showed essentially the same results.

A Cox proportional hazards analysis (Table 4) showed that the multivariate HR (95% CI) for drinking lapse was 0.63 (0.42–0.95) in nonsmokers before and at the time of either the first drinking lapse or the end of follow-up without a lapse, compared with smokers, while the HR for heavy-drinking lapse was 0.58 (0.37–0.91).

Higher daily alcohol intake and taking psychiatric medications at the time of discharge were associated with higher HRs for both drinking lapse and heavy-drinking lapse. Family history of heavy or problem drinking also increased the HR for drinking lapse.

Table 5 shows the smoking status in the 128 baseline smokers before and after the program. The mean daily number of cigarettes decreased from 19.2±10.2 before the program to 8.8±8.3 during the first month after the program. The nonsmoking rate during the first month was 28.9%. The improvement in the smoking status remained significant on the last follow-up letters, though a slight return toward the smoking status seen before the program was observed.

## Discussion

The present prospective study demonstrated that the cigarette smoking status was strongly associated with the drinking outcome after a 3-month inpatient ARP with simultaneous smoking cessation program. Nonsmokers after discharge showed much better outcomes for alcohol abstinence and heavy-drinking-free status throughout a 12-month period, regardless of the baseline smoking status. Smoking cessation interventions for people with psychiatric disorders have been shown to be effective, but such interventions are more difficult than those for people

**Table 4. Drinking status during 12 months after inpatient program and its determinants including smoking status in Japanese alcohol-dependent men.**

| | Drinking relapse | Heavy drinking |
|---|---|---|
| | Multivariate HR (95% CI)* | Multivariate HR (95% CI)* |
| Age (per +10 yrs) | 1.10 (0.92–1.31) | 1.08 (0.88–1.32) |
| Age at first drinking (yrs) | | |
| ≤15 yrs | 1.20 (0.65–2.22) | 1.62 (0.81–3.27) |
| 16–19 yrs | 0.95 (0.58–1.56) | 1.25 (0.71–2.21) |
| ≥20 yrs | referent | referent |
| Age at the start of regular drinking (yrs) | | |
| ≤19 yrs | 1.05 (0.58–1.91) | 1.14 (0.58–2.22) |
| 20–25 yrs | 1.34 (0.83–2.16) | 1.58 (0.90–2.76) |
| ≥26 yrs | referent | referent |
| Usual alcohol intake (per +22 g/day) | 1.09 (1.02–1.17) | 1.10 (1.02–1.18) |
| Alcoholic beverage most frequently consumed | | |
| beer/canned chuhai (4–9%) | 1.06 (0.67–1.68) | 0.88 (0.52–1.50) |
| sake/wine (12–16%) | 1.47 (0.85–2.54) | 1.56 (0.86–2.84) |
| shochu/whiskey/other spirits (20–40%) | referent | referent |
| Non-smoking before and at the time of drinking relapse or censoring | 0.63 (0.42–0.95) | 0.58 (0.37–0.91) |
| Living alone | 0.87 (0.57–1.33) | 0.97 (0.61–1.54) |
| Unemployed | 1.15 (0.76–1.75) | 1.27 (0.80–2.02) |
| Family history of heavy/problem drinking | 1.53 (1.03–2.25) | 1.31 (0.86–2.02) |
| Lifetime episodes of other psychiatric disorders | 1.08 (0.67–1.73) | 1.14 (0.67–1.92) |
| Medication at discharge | | |
| Acamprosate or disulfiram | 1.00 (0.67–1.51) | 1.00 (0.64–1.56) |
| Other psychiatric medication | 1.64 (1.09–2.46) | 1.67 (1.06–2.63) |

Multivariate hazard ratio (HR) and 95% confidence interval (CI) by the Cox proportional hazards model, where all variables were simultaneously entered into the model

without a history of psychiatric disorders [13]. After adjusting for age, drinking and smoking habits, lifestyle, family history of heavy or problem drinking, lifetime episodes of other major psychiatric disorders, and medications at discharge, the multivariate HRs for drinking lapse and a heavy-drinking lapse during the 12-month period were 0.41–0.63 and 0.47–0.58, respectively, for the nonsmoking after discharge group.

**Table 5. Smoking status before and after a 3-month inpatient program for simultaneous alcohol abstinence and smoking cessation in 128 Japanese alcohol-dependent male smokers.**

| | Before the program | | First month of follow-up | | Last follow-up letter | |
|---|---|---|---|---|---|---|
| | N | (%) | N | (%) | N | (%) |
| Cigarette smoking | | | | | | |
| None | 0 | (0.0%) | 37 | (28.9%) | 32 | (25.0%) |
| 1–19 cigs/d | 51 | (39.8%) | 62 | (48.4%) | 47 | (36.7%) |
| 20+ cigs/d | 77 | (60.2%) | 27 | (21.1%) | 47 | (36.7%) |
| 1+, unknown | 0 | (0.0%) | 2 | (1.6%) | 2 | (1.6%) |
| p vs before the program** | | | | <0.0001 | | <0.0001 |
| mean±SD (cigs/d) | 19.2±10.2 | | 8.8±8.3* | | 12.2±9.6* | |
| p vs before the program** | | | | <0.0001 | | <0.0001 |

* Excluding subjects who reported smoking but not the number of cigarettes that were smoked.

** Wilcoxon signed-rank test.

Alcohol-dependent individuals and treatment providers have often shared beliefs such as "It is more difficult to establish alcohol abstinence and smoking cessation simultaneously than to achieve alcohol abstinence alone," "Smoking can be useful to suppress drinking urges" or "Smoking is less harmful than drinking in subjects with alcohol dependence." These beliefs partly explain the unwillingness to introduce smoking cessation programs in treatment settings for alcohol dependence [1, 3]. However, research has revealed that nonsmoking or interventions for smoking cessation enhance, rather than compromise, alcohol abstinence in alcohol-dependent individuals [2, 3, 7, 8]. Continued smoking and smoking initiation among nonsmokers increased the risk of alcohol and other drug use disorder relapse [14]. The present study using a prospective follow-up design clearly supports these previous findings for the first time in an Asian population.

Nicotine dependence and alcohol and other substance dependence share common biopsychosocial/environmental backgrounds [3], and human laboratory studies have shown that alcohol and tobacco have reciprocal influences on potentiating craving: alcohol increases the craving to smoke, and tobacco and nicotine increase alcohol cravings [15]. Among heavy drinkers participating in smoking cessation treatment, even moderate alcohol use was associated with an increased risk of smoking [16]. Nicotine enhances the effects of other substances as a gateway drug [17]. Alcohol-dependent never/ex-smokers showed better neurocognitive recovery during alcohol abstinence than alcohol-dependent smokers [18]. A meta-analysis of 26 longitudinal studies assessing mental health showed that smoking cessation is associated with reduced depression, anxiety, and stress and an improved positive mood and quality of life [19]. Smoking is associated with mental distress, quality of life and treatment drop-out among patients receiving primary alcohol use disorder treatment [20]. The improved recovery of neurocognitive stability partly accounts for the better drinking status outcomes in nonsmoking alcohol-dependent subjects. Collectively, these findings partly explain why a smoking cessation program enhances or contributes to alcohol abstinence.

Various health risks are exacerbated by concurrent alcohol-dependence and smoking [4, 6, 21]. Long-term follow-up after a program for alcohol or other non-nicotine drug dependence showed that more individuals died from tobacco-related diseases than from alcohol-related causes [5]. The proportion of current smokers in the presently reported population was 64.6%, and 60.2% of them smoked 20+ cigarettes/day. This current smoking rate is double the rate for Japanese adult men (32.2%) according to the 2014 National Health and Nutrition Survey [22]. Out of the baseline smokers, the nonsmoking rate increased to 28.9% and the heavy smoking rate of 20+ cigarettes/day decreased to 21.1% during the first month after discharge. These rates were 25.0% and 36.7%, respectively, at the time of the last follow-up questionnaire. Although maintaining smoking cessation for a long term is difficult in alcohol-dependent smokers [23], our inpatient program achieved positive results with regard to smoking status, and has probably contributed to a reduction in harm from smoking-related comorbidity.

Our earlier study showed that many alcohol-dependent subjects were highly motivated regarding both smoking cessation and alcohol abstinence during the inpatient program [9]. The subjects' scores on a 0- to 10-point scale for rating motivation for smoking cessation showed that 40.1% and 23.9% of them reported scores of 10 and 8–9, respectively, after a 3-week hospital stay. Research has shown that a large majority of smokers with alcohol use disorders wish to quit smoking [24, 25], and recent general population trends toward nonsmoking have become evident among persons with alcohol and other drug problems [26]. This trend has also been observed among Japanese alcohol-dependent men (aged 40–79, n = 7545): the current smoking rates were 86.7%, 82.2%, 79.9%, 75.3%, and 67.5% during the periods of 1993–1997, 1998–2002, 2003–2007, 2008–2012, and 2013–2018, respectively [21]. Thus, there is a growing demand for smoking cessation programs in treatment settings for alcohol dependence.

Our inpatient program had several unique points. A 3-month inpatient program is provided as a standard ARP by Japanese universal health insurance. A smoking cessation program was integrated into the ARP because of an institutional policy of no smoking on the premises. The implementation of a tobacco-free grounds policy is associated with a lower prevalence of smoking in residential substance use disorder treatment programs [27]. The efficacy of the smoking-cessation agent varenicline has been reported in Japanese smokers [28]. The efficacy of varenicline has been shown to be superior to that of nicotine patches in both cohorts with and without psychiatric disorders [29], and we did not offer nicotine replacement therapy. We recommended current smokers to use varenicline, and about half of the current smokers took varenicline during the 3-month hospital stay. In our previous report [9], heavier smokers in an alcohol-dependent smoking inpatients were more likely to choose the use of varenicline, and the success rate of varenicline users in quitting smoking during hospitalization was significantly higher than that of non-users. In this study, 70.5% of varenicline users smoked 20 or more cigarettes, more frequently than 50.7% of non-users (p = 0.030), and the multivariate OR for smoking cessation within 1 month of discharge was 2.50 (1.00–6.25), indicating that in baseline smokers varenicline use was the only predictor of smoking cessation, along with abstinence from alcohol (multivariate OR = 2.77 [1.01–7.61]). The incidence of adverse effects from varenicline, especially nausea, is relatively high in general populations, but varenicline is well-tolerated among alcohol-dependent smokers [9, 30]. A meta-analysis of randomized controlled trials showed that varenicline may promote smoking cessation in people with alcohol dependence [31]. In addition, varenicline has been reported to reduce alcohol consumption and cravings [31–33]. Four study meetings for smoking cessation were held. The subjects received medical examinations and were encouraged to stop smoking by both physicians and psychiatrists. The enhanced impact of smoking cessation on long-term alcohol abstinence as well as mental and physical recovery was always highlighted.

Previous studies have shown that family history of alcohol dependence was associated with a more recurrent course of alcohol dependence [34, 35]. The results of the present study also showed a higher HR for drinking lapse in Japanese alcohol-dependent population with family history of heavy or problem drinking. Some genetic and environmental backgrounds associated with the family history may influence the severity of alcohol dependence [34]. The use of psychiatric medications at discharge, primarily benzodiazepines as sleep medications and antidepressants, was associated with worse drinking outcomes. This may reflect the fact that patients on medications continued to have more severe sleep disturbances and depression. According to one review, benzodiazepine receptor agonists for sleep are widely avoided by addiction medicine physicians in the United States after the alcohol withdrawal period (3–7 days) because of their abuse potential and overdose risk when taken with alcohol [36]. However, long-term use of benzodiazepines for insomnia in alcohol-dependent patients was common in our Center at the time, with about half of the patients using benzodiazepines as a sleep medication upon discharge. The easy administration of benzodiazepines itself may have influenced the worse prognosis for sobriety, and we now use behavioral therapies and, if necessary, orexin receptor antagonists as first-line agents for insomnia in alcohol-dependent patients.

The present study had several limitations. First, this study was not designed to examine the effect of a smoking cessation program concurrent to alcohol abstinence treatment on the smoking and drinking status outcomes, since control groups that did not undergo the smoking cessation program were not included in the study. Therefore, we cannot formally evaluate the effect of the smoking cessation program on the smoking status, although according to the first month follow-up and the last follow-up letter, 25%-29% of the baseline smokers quit smoking and the average number of cigarettes smoked decreased. Drinking and smoking status were inquired using self-reported questionnaire without the objective measurement of alcohol and smoking cessation

(e.g., CO measurement or collateral reports). The reliability of self-reported drinking and smoking statuses is always problematic among alcohol-dependent subjects. The subjects were assured that their answers would be kept confidential and would not be shared with concerned medical staff. We cannot rule out the possibility that the Kaplan-Meier curves for the 198 follow-up subjects overestimated the rates of alcohol abstinence, since 18 subjects dropped out because of interruptions in their replies. However, a sensitivity analysis performed with the assumption that all the dropout subjects drank or drank heavily at the time of dropout showed essentially unchanged or enhanced results supporting better outcomes for alcohol abstinence and heavy-drinking-free status in nonsmoking groups. The analysis sample of 198 first-time treatment patients was highly selective because patients who received multiple treatments were excluded due to their very poor sobriety prognosis. Therefore, a sensitivity analysis was conducted using a total of 273 patients, including those who had received multiple treatments. The results showed that the abstinence rate decreased, but the association between smoking status and abstinence did not change. To adequately answer the question whether the smoking cessation program would positively impact alcohol abstinence rates, we would need a control group without smoking cessation treatment. Other confounding variables that were not measured or examined (e.g., motivation for change, resilience, environmental cues, addiction/relapse risk factors), rather than the treatment effect, might explain the association between smoking and alcohol cessation. It's possible, for example, that refraining from smoking may limit exposure to environmental cues that can cause smoking and drinking. Therefore, it could be the change in exposure to these cues that is largely driving the alcohol abstinence rates, rather than the smoking status on its own. Our earlier study showed that the patients' scores for motivation and self-efficacy for smoking cessation were significantly correlated with their scores for motivation and self-efficacy for alcohol abstinence [9]. It is possible that those who were able to quit smoking were also highly motivated and resilient in their sobriety. Also, the 9 letters and telephone encouragement *per se* might have contributed to the maintenance of alcohol abstinence and smoking cessation. These confounding factors limit what we can causally attribute to the treatment program. Out of the 242 first-time treatment patients, 44 (18.2%) were not followed and that may have inflated the abstinence rates in this study, although there were no significant differences in their background characteristics except for age. Finally, the generalization of the results obtained in the present study examining treatment-seeking alcohol-dependent men will require further confirmation in various high-risk drinking populations, including patients with milder alcohol use disorders and female patients.

In conclusion, the present prospective follow-up study demonstrated that nonsmoking was associated with better outcomes on the drinking status of Japanese alcohol-dependent men after a 3-month inpatient program for simultaneous alcohol abstinence and smoking cessation. Smoking cessation programs may be recommended to be integrated into alcohol abstinence programs.

## Supporting information

**S1 Dataset.**
(XLSX)

**S1 Fig. Smoking status within 1 month after discharge and drinking status after a 3-month inpatient program for alcohol abstinence and smoking cessation in Japanese alcohol-dependent men.** The sensitivity analysis for Fig 1 with the assumption that all the dropout subjects drank (left) or drank heavily (right) at the time of dropout showed essentially unchanged or enhanced results supporting better outcomes for alcohol abstinence and heavy-drinking-free status in nonsmoking groups.
(TIF)

**S2 Fig. Smoking status within 1 month after discharge and drinking status after a 3-month inpatient program for alcohol abstinence and smoking cessation in Japanese alcohol-dependent men.** The sensitivity analysis for Fig 1 using the 272 first-time and multiple treatment patients showed essentially unchanged results supporting better outcomes for alcohol abstinence and heavy-drinking-free status in nonsmoking groups.
(TIF)

**S3 Fig. Smoking status and drinking status after a 3-month inpatient program for alcohol abstinence and smoking cessation in Japanese alcohol-dependent men.** The sensitivity analysis for Fig 2 with the assumption that all the dropout subjects drank (left) or drank heavily (right) at the time of dropout showed essentially unchanged or enhanced results supporting better outcomes for alcohol abstinence and heavy-drinking-free status in nonsmoking groups.
(TIF)

**S4 Fig. Smoking status and drinking status after a 3-month inpatient program for alcohol abstinence and smoking cessation in Japanese alcohol-dependent men.** The sensitivity analysis for Fig 2 using the 272 first-time and multiple treatment patients showed essentially unchanged results supporting better outcomes for alcohol abstinence and heavy-drinking-free status in nonsmoking groups.
(TIF)

## Author Contributions

**Conceptualization:** Akira Yokoyama, Mitsuru Kimura, Atsushi Yoshimura, Sachio Matsushita, Susumu Higuchi.

**Data curation:** Akira Yokoyama, Mitsuru Kimura, Atsushi Yoshimura.

**Formal analysis:** Akira Yokoyama, Tetsuji Yokoyama.

**Funding acquisition:** Susumu Higuchi.

**Investigation:** Mitsuru Kimura, Atsushi Yoshimura, Junichi Yoneda, Hitoshi Maesato, Yasunobu Komoto, Hideki Nakayama, Hiroshi Sakuma, Yosuke Yumoto, Tsuyoshi Takimura, Tomomi Toyama, Chie Iwahara, Takeshi Mizukami.

**Methodology:** Akira Yokoyama, Mitsuru Kimura, Atsushi Yoshimura.

**Project administration:** Atsushi Yoshimura.

**Writing – original draft:** Akira Yokoyama, Tetsuji Yokoyama.

**Writing – review & editing:** Akira Yokoyama, Mitsuru Kimura, Atsushi Yoshimura, Sachio Matsushita, Tetsuji Yokoyama.

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
