## [Decision Letter · Decision Letter 0]

27 Jan 2023

PONE-D-22-26289Nonsmoking after simultaneous alcohol abstinence and smoking cessation program yielded better drinking status outcome in Japanese alcohol-dependent men: a prospective follow-up studyPLOS ONE

Dear Dr. Yokoyama,

Thank you for submitting your manuscript to PLOS ONE. After careful consideration, we feel that it has merit but does not fully meet PLOS ONE’s publication criteria as it currently stands. Therefore, we invite you to submit a revised version of the manuscript that addresses the points raised during the review process.

We look forward to receiving your revised manuscript.

Kind regards,

Billy Morara Tsima, MD MSc

Academic Editor

PLOS ONE

and https://journals.plos.org/plosone/s/file?id=ba62/PLOSOne_formatting_sample_title_authors_affiliations.pdf.

“This research was supported by Health Labour Sciences Research Grant, Japan (H26-seishin-ippan-006) and Intramural Research Grant for Neurological and Psychiatric Disorders of National Center of Neurology and Psychiatry (25-2).”

3. Thank you for stating the following in the Acknowledgments/ Funding Section of your manuscript:

“This research was supported by Health Labour Sciences Research Grant, Japan (H26-seishin-ippan-006) and Intramural Research Grant for Neurological and Psychiatric Disorders of National Center of Neurology and Psychiatry (25-2).”

“This research was supported by Health Labour Sciences Research Grant, Japan (H26-seishin-ippan-006) and Intramural Research Grant for Neurological and Psychiatric Disorders of National Center of Neurology and Psychiatry (25-2).”

Reviewers' comments:

Reviewer's Responses to Questions

**Comments to the Author**

1. Is the manuscript technically sound, and do the data support the conclusions?

Reviewer #1: Partly

Reviewer #2: Yes

2. Has the statistical analysis been performed appropriately and rigorously? 

Reviewer #1: I Don't Know

Reviewer #2: Yes

3. Have the authors made all data underlying the findings in their manuscript fully available?

Reviewer #1: No

Reviewer #2: Yes

4. Is the manuscript presented in an intelligible fashion and written in standard English?

Reviewer #1: Yes

Reviewer #2: Yes

5. Review Comments to the Author

Reviewer #1: This manuscript presents results from a prospective study of 198 Japanese alcohol-dependent men who completed a simultaneous alcohol abstinence and smoking cessation program. Through survival analysis, the authors sought to examine the effect of smoking status on alcohol abstinence and, based on those findings, extrapolate whether a smoking cessation program should be recommended as part of or integrated into alcohol abstinence programs. The authors found that hazard ratios for drinking and heavy-drinking relapse tended to be noticeably lower among non-smokers than smokers. As such, the authors further concluded that, "Smoking cessation programs may be recommended to be integrated into alcohol abstinence programs," (pg. 34). The manuscript generally well-written, and the analyses are well-documented. However, there are some major concerns with this study two of which are documented below:

1) There are many concerns with the study sample. First, the analytic sample of 198 participants is generally considered small for a survival analysis. Second, the authors excluded 112 participants who required multiple rounds of alcohol abstinence treatment from the analytic sample due to the prognosis of sobriety being very poor for patients requiring multiple treatments (pg. 17). It is unclear what this has to do with the effect of concurrent smoking cessation treatment on alcohol abstinence, and applying this exclusion criteria yields a highly selective analytic sample. The authors should have done a sensitivity analyses in this subpopulation.

2) This study is not designed to examine the effect of a smoking cessation program concurrent to alcohol abstinence treatment on alcohol abstinence outcomes. The authors take smoking status as a proxy for the effect of the smoking cessation program potentially without formally evaluating the effect of the smoking cessation program on smoking status. The closest the authors come to doing this is comparing the frequency distribution of cigarette smoking before the treatment program to the first month of follow-up and the last follow-up letter. Additionally, the authors do not consider alternate explanations for the relationship between smoking status and alcohol abstinence. It's possible, for example, that refraining from smoking may limit exposure to environment cues that can cause smoking and drinking. Therefore, it would be the change in exposure to these cues that is largely driving the abstinence rates rather than smoking status on its own. Ultimately, the authors have not fully considered all sources of confounding in their study, and this limits what they can causally attribute to the treatment program.

Reviewer #2: The manuscript presents an interesting work which aimed to explore if intervention for smoking cessation could enhance alcohol abstinence in treatment settings for alcohol dependence in Asians. It was found that nonsmoking had a better impact on the drinking status of Japanese alcohol-dependent men. The authors recommended that a smoking cessation program may be integrated into alcohol abstinence programs. In my opinion, by the data and results, this manuscript can be accepted for publication in PLOS ONE.

6. PLOS authors have the option to publish the peer review history of their article (what does this mean?). If published, this will include your full peer review and any attached files.

Reviewer #1: No

Reviewer #2: No

---

## [Author Response · Author response to Decision Letter 0]

20 Feb 2023

Point by point reply

Manuscript ID PONE-D-22-26289

Nonsmoking after simultaneous alcohol abstinence and smoking cessation program was associated with better drinking status outcome in Japanese alcohol-dependent men: a prospective follow-up study

Dear Professor Billy Morara Tsima

Academic Editor

PLOS ONE

We appreciate the comments of you and two reviewers. They were helpful in strengthening and clarifying portions of the manuscript. 

We ensure that our manuscript meets PLOS ONE's style requirements, and added the following information in the cover letter: This research was supported by Health Labour Sciences Research Grant, Japan (H26-seishin-ippan-006) and Intramural Research Grant for Neurological and Psychiatric Disorders of National Center of Neurology and Psychiatry (25-2). The funders had no role in study design, data collection and analysis, decision to publish, or preparation of the manuscript. In addition, we uploaded our study’s minimal underlying data set as Supporting Information files.

Reply to Reviewers' comments:

Reviewer #1: This manuscript presents results from a prospective study of 198 Japanese alcohol-dependent men who completed a simultaneous alcohol abstinence and smoking cessation program. Through survival analysis, the authors sought to examine the effect of smoking status on alcohol abstinence and, based on those findings, extrapolate whether a smoking cessation program should be recommended as part of or integrated into alcohol abstinence programs. The authors found that hazard ratios for drinking and heavy-drinking relapse tended to be noticeably lower among non-smokers than smokers. As such, the authors further concluded that, "Smoking cessation programs may be recommended to be integrated into alcohol abstinence programs," (pg. 34). The manuscript generally well-written, and the analyses are well-documented. However, there are some major concerns with this study two of which are documented below:

1) There are many concerns with the study sample. First, the analytic sample of 198 participants is generally considered small for a survival analysis. Second, the authors excluded 112 participants who required multiple rounds of alcohol abstinence treatment from the analytic sample due to the prognosis of sobriety being very poor for patients requiring multiple treatments (pg. 17). It is unclear what this has to do with the effect of concurrent smoking cessation treatment on alcohol abstinence, and applying this exclusion criteria yields a highly selective analytic sample. The authors should have done a sensitivity analyses in this subpopulation.

Reply: According to the reviewer’s comment, we performed sensitivity analyses for Figures 1 and 2 using a total of 272 patients, including first-time and multiple treatment patients; the results did not change. 

Results: Patients receiving multiple treatments were excluded due to their very poor sobriety prognosis. Of these, 74 responded to the questionnaire, and their 1-year abstinence rate was 23.1% ± 5.3% (mean ± SE), while the abstinence rate for the 198 first-time treatment patients was 41.0% ± 3.6%. A sensitivity analysis was also performed for Fig. 1 using a total of 272 first-time and multiple treatment patients (Fig. S2). Although the overall abstinence rate decreased, the results of the sensitivity analysis were essentially unchanged.

Another sensitivity analysis for Fig. 2 was performed using the 272 first-time and multiple treatment patients described above (Fig. S4). Although the overall abstinence rate decreased, the sensitivity analysis showed essentially the same results.

Discussion (Limitations): The analysis sample of 198 first-time treatment patients was highly selective because patients who received multiple treatments were excluded due to their very poor sobriety prognosis. Therefore, a sensitivity analysis was conducted using a total of 273 patients, including those who had received multiple treatments. The results showed that the abstinence rate decreased, but the association between smoking status and abstinence did not change.

2) This study is not designed to examine the effect of a smoking cessation program concurrent to alcohol abstinence treatment on alcohol abstinence outcomes. The authors take smoking status as a proxy for the effect of the smoking cessation program potentially without formally evaluating the effect of the smoking cessation program on smoking status. The closest the authors come to doing this is comparing the frequency distribution of cigarette smoking before the treatment program to the first month of follow-up and the last follow-up letter. Additionally, the authors do not consider alternate explanations for the relationship between smoking status and alcohol abstinence. It's possible, for example, that refraining from smoking may limit exposure to environment cues that can cause smoking and drinking. Therefore, it would be the change in exposure to these cues that is largely driving the abstinence rates rather than smoking status on its own. Ultimately, the authors have not fully considered all sources of confounding in their study, and this limits what they can causally attribute to the treatment program.

Reply: According to this comment, descriptions suggesting an influence of smoking cessation on alcohol abstinence, such as “yielded” and “affected,” have been changed to 'is associated with' and other wordings indicating an association only. We have also added the following sentences to the section on study limitations. 

Title: Nonsmoking after simultaneous alcohol abstinence and a smoking cessation program was associated with a better drinking status outcome in Japanese alcohol-dependent men: a prospective follow-up study

Discussion: The present study had several limitations. First, this study was not designed to examine the effect of a smoking cessation program concurrent to alcohol abstinence treatment on the smoking and drinking status outcomes, since control groups that did not undergo the smoking cessation program were not included in the study. Therefore, we cannot formally evaluate the effect of the smoking cessation program on the smoking status, although according to the first month follow-up and the last follow-up letter, 25%-29% of the baseline smokers quit smoking and the average number of cigarettes smoked decreased.

Other confounding variables that were not measured or examined (e.g., motivation for change, resilience, environmental cues, addiction/relapse risk factors), rather than the treatment effect, might explain the association between smoking and alcohol cessation. It's possible, for example, that refraining from smoking may limit exposure to environmental cues that can cause smoking and drinking. Therefore, it could be the change in exposure to these cues that is largely driving the alcohol abstinence rates, rather than the smoking status on its own. Our earlier study showed that the patients’ scores for motivation and self-efficacy for smoking cessation were significantly correlated with their scores for motivation and self-efficacy for alcohol abstinence [9]. It is possible that those who were able to quit smoking were also highly motivated and resilient in their sobriety. Also, the 9 letters and telephone encouragement per se might have contributed to the maintenance of alcohol abstinence and smoking cessation. These confounding factors limit what we can causally attribute to the treatment program.

Another change: A coauthor suggested that a “family history of alcohol dependence” might be inappropriate; although some diagnoses might have been made by physicians, many seem to have been based on family impressions. In our questionnaire, the items asking about a family history of problem drinking included four categories: heavy drinking, problem drinking, alcohol dependence, and death due to drinking. We changed “family history of alcohol dependence” to an overall category of “family history of heavy or problem drinking among grandparents, parents, and siblings,” and conducted analyses using this category. The results essentially did not change.

Reviewer #2: The manuscript presents an interesting work which aimed to explore if intervention for smoking cessation could enhance alcohol abstinence in treatment settings for alcohol dependence in Asians. It was found that nonsmoking had a better impact on the drinking status of Japanese alcohol-dependent men. The authors recommended that a smoking cessation program may be integrated into alcohol abstinence programs. In my opinion, by the data and results, this manuscript can be accepted for publication in PLOS ONE.

Reply: Thank you. A coauthor suggested that a “family history of alcohol dependence” might be inappropriate; although some diagnoses might have been made by physicians, many seem to have been based on family impressions. In our questionnaire, the items asking about a family history of problem drinking included four categories: heavy drinking, problem drinking, alcohol dependence, and death due to drinking. We changed “family history of alcohol dependence” to an overall category of “family history of heavy or problem drinking among grandparents, parents, and siblings,” and conducted analyses using this category. The results essentially did not change.

---

## [Editor Report · Decision Letter 1]

1 Mar 2023

Nonsmoking after simultaneous alcohol abstinence and smoking cessation program was associated with better drinking status outcome in Japanese alcohol-dependent men: a prospective follow-up study

PONE-D-22-26289R1

Dear Dr. Yokoyama,

We’re pleased to inform you that your manuscript has been judged scientifically suitable for publication and will be formally accepted for publication once it meets all outstanding technical requirements.

Kind regards,

Billy Morara Tsima, MD MSc

Academic Editor

PLOS ONE
---

## [Editor Report · Acceptance letter]

3 Mar 2023

PONE-D-22-26289R1 

Nonsmoking after simultaneous alcohol abstinence and smoking cessation program was associated with better drinking status outcome in Japanese alcohol-dependent men: a prospective follow-up study 

Dear Dr. Yokoyama:

I'm pleased to inform you that your manuscript has been deemed suitable for publication in PLOS ONE. Congratulations! Your manuscript is now with our production department. 

Kind regards, 

on behalf of

Dr. Billy Morara Tsima 

Academic Editor

PLOS ONE